# Chaotic Transformers for Deep Reinforcement Learning in Algorithmic Trading

## Abstract

Chaotic Transformers for Deep Reinforcement Learning can be applied in algorithmic trading to improve the efficiency and effectiveness of trading strategies. In algorithmic trading, deep reinforcement learning can be used to learn trading policies that maximize the expected reward. However, due to the highly complex and nonlinear nature of financial markets, it can be challenging to identify profitable trading opportunities and avoid overfitting to historical data.

## 1 Introduction

The paper proposes a novel approach to Algorithmic trading by integrating three powerful concepts - chaotic neural networks, transformers, and deep reinforcement learning (DRL). While DRL has been used in several works on Algorithmic trading, the use of transformers in this domain is relatively unexplored. Transformers are among the most robust models in deep learning and their incorporation can significantly enhance the performance of the trading system (Liu et al., 2022; Wu et al., 2021).

Moreover, chaotic neural networks can add value by modeling the stochastic nature of the stock markets, which is highly relevant given the volatile nature of financial markets. Therefore, this paper seeks to leverage the strengths of these three concepts to develop an improved Algorithmic trading system (Wang & Lee, 2021; Ning et al., 2009).

## 2 Related Work

In (Wang & Lee, 2021) Chaotic recurrent neural network (CRNN) which uses chaotic neural network, Lee-Oscillator, as activation function, instead of traditional sigmoid-based activation function, has been introduced.

The (Carta et al., 2021) proposes an ensemble of Deep Q-learning classifiers with different experiences in the market for intraday stock trading. The approach is flexible, with diverse combinations of actions and an ensemble strategy done with different agreement thresholds. The experiments show promising results, especially with the Only-Long agent for markets with a typical upward behavior and the Long+Short and Full agents for better mitigation of negative market periods. The limitation of the approach is that it only processes past prices data, and future work could include coupling the system with news-based classification systems, validation on other types of markets, experimenting with different types of agents and network architectures, and exploring other hyperparameters.

The (Liu et al., 2022) proposes a new approach to time series forecasting that increases series stationarity while still incorporating non-stationary information, improving data predictability and model predictive capability. The proposed Non-stationary Transformers framework is tested on six real-world benchmarks, and detailed derivations and ablations are provided to show the effectiveness of each component.

## 3 Methodology

Our approach is based on all three stated papers, plus some extra techniques which have been used in other papers. We use a Multi-DQN RL system which uses a non-stationary transformer with Chaotic activation. The stock trend has been detected in the first part and the size and price of the trade are

calculated based on the output probability of the model using common techniques in Algorithmic Trading. Also, we use denoising on the input historical price data which has been used (Park & Lee, 2021).

## 4 CONCLUSION

In this paper, we proposed a mixture of an ensemble DRL model with a Transformer based neural network with Chaotic activation. Also, denoising as a useful preprocessing technique has been used. The model first predicts the stock market's future trend and then opens a position based on the predictions to get acceptable profit.

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
