# OpenReview forum: "Chaotic Transformers for Deep Reinforcement Learning in Algorithmic Trading"
_ICLR.cc/2023/TinyPapers — Submitted to Tiny Papers @ ICLR 2023_

### Official Review · Reviewer_EdJc · 2023-03-22

**Confidence:** 4

**Summary Of Contributions:**

The paper considers identifying profitable trading strategies by leveraging chaotic transformers to train on historical financial data while avoid overfitting.

**Rating:**

Needs Clarification (NC): a submission which does not meet the reviewing criteria and needs clarification for its described problem or solution

**Strengths And Weaknesses:**

The article proposes the use of Chaotic Transformers for identifying profitable trading strategies that could avoid overfitting.

Strengths:
- The authors propose the combination of (1) the chaotic activation function, (2) the transformers architecture, (3) deep reinforcement learning-based trading algorithms, (4) newer datasets for training, and (5) developments in time-series data. Given that these methods and ideas perform well individually, the authors believe that combining these approaches can lead to a superior and profitable trading strategy.

Weaknesses:
- The article lacks significant details about the proposed algorithm. In particular, exact system diagrams, experiments as well as comparisons with baselines are missing.
- Relevant literature analysis and intuition to build the new diagram are missing as well.
- Overall, without these details, it is hard to evaluate the proposed technique.

**Suggested Changes:**

- The article proposes a new idea and architecture. However, it does not include the exact details about how the various components are being mixed together. Due to the lack of information about how these components interact, it is hard to evaluate the newly proposed approach. Adding a system diagram will aid in understanding the direct contribution of this paper.
- Experimentation and comparisons are required to showcase the efficacy of the newly developed trading strategies. Baselines as well as relevant hyperparameter tuning will aid in understanding the methodology better. The computational cost of the new algorithm should be included as well.
- Relevant literature to support the development of such an algorithm should be added. It could provide the motivation for studying the new algorithm.

---

### Official Review · Reviewer_BQmL · 2023-03-30

**Confidence:** 3

**Summary Of Contributions:**

Authors have given an idea to of using chaotic neural networks, transformers, and deep reinforcement learning (DRL) to learn trading policies.

**Rating:**

Needs Clarification (NC): a submission which does not meet the reviewing criteria and needs clarification for its described problem or solution

**Strengths And Weaknesses:**

Strong points:
1. Authors idea is to merge DRL, transformers and chaotic activations in predicting stocks as well its trade size and price.\

Weakness:
1. There is no proper block diagram have been shown how the authors are planning to achieve their objective.
2. Idea seems interesting but there is nothing has been mentioned about data, experimental setup, codes locations, which makes it difficult to understand how the idea can be reproduced by the others as well as what are the findings user themselves after experimenting.


**Suggested Changes:**

1. Please add the URM statement before references.
2. As idea seems to be novel but there is no proof has been given to verify its validity. Therefore, authors can give first a small block diagram to explain the components of their idea in brief. Later on, authors can explain the findings of their results obtained using their methods through one figure or table.
3. Finally authors should share the data as well as code repository for other users to reproduce the results.

---

### Comment · Area_Chair_dKGK · 2023-06-06
**Meta-review: Do not invite to archive**

Unfortunately the authors have not responded to any of the points raised by myself or the reviewers, nor have they adjusted the paper in any meaningful way. Therefore I am sorry to say I cannot change my decision and __do not recommend this paper for archive__.

---

### Meta-Review · Area_Chair_dKGK · 2023-04-04

**Recommendation:** Invite to revise
**Confidence:** 4

**Metareview:**

In this paper the authors have alluded to the use of chaotic neural networks, transformers and deep reinforcement learning to learn trading policies. Unfortunately the reviewers are both in agreement that no substantial details about the proposed algorithm, system diagram, experiments or results are provided. Likewise, there are no comparisons with baselines. It is currently unclear whether the authors have not yet run any experiments or have run experiments but are withholding details. Relevant literature background is not provided and some grammatical errors make the text hard to understand in places.

Whilst we believe in principle that the authors model could hold promise, in its current form we cannot recommend the paper for archiving.
The authors have only used one of their two available pages, so there exists space to add more details outlining their contribution.

To be clear, we recommend that, at a minimum, the authors should:
* Give a clear block diagram or description of their algorithm.
* Describe some experiments and their results.
* Compare these results to relevant baselines or other comparable models.
* Add a few sentences describing the relevant literature motivating the algorithm they are proposing.

**Summary:**

In this paper the authors have alluded to the use of chaotic neural networks, transformers and deep reinforcement learning to learn trading policies. Whilst the authors idea could, in principle, be interesting, the reviewers agree that no substantial details about the methodology or results are provided, making it incredibly hard to evaluate their contribution.

**Comments And Feedback To The Authors:**

We recommend the authors should use the remaining space to:

* Give a clear block diagram or description of their algorithm.
* Describe some experiments and their results.
* Compare these results to relevant baselines or other comparable models.
* Add a few sentences describing the relevant literature motivating the algorithm they are proposing.

**Reason For Not Giving A Higher Recommendation:**

The reviewers agree that it is very difficult to evaluate the model (which is sparsely described) or the results (which are not described).

**Reason For Not Giving A Lower Recommendation:**

N/A

---

### Decision · Program_Chairs · 2023-04-08

No revision received; not invited to archive